# Learning generative models for demixing of structured signals from their superposition using GANs

## ABSTRACT

Recently, Generative Adversarial Networks (GANs) have emerged as a popular alternative for modeling complex high dimensional distributions. Most of the existing works implicitly assume that the clean samples from the target distribution are easily available. However, in many applications, this assumption is violated. In this paper, we consider the problem of learning GANs under the observation setting when the samples from target distribution are given by the superposition of two structured components. We propose two novel frameworks: denoising-GAN and demixing-GAN. The denoising-GAN assumes access to clean samples from the second component and try to learn the other distribution, whereas demixing-GAN learns the distribution of the components at the same time. Through comprehensive numerical experiments, we demonstrate that proposed frameworks can generate clean samples from unknown distributions, and provide competitive performance in tasks such as denoising, demixing, and compressive sensing.

## 1 INTRODUCTION

In this paper, we consider the classical problem of separating two structured signals observed under the following superposition model:

$$Y = X + N, \tag{1}$$

where $X \in \mathcal{X}$ and $N \in \mathcal{N}$ are the constituent signals, and $\mathcal{X}, \mathcal{N} \subseteq \mathbb{R}^p$ denote the two structured sets. In general the separation problem is inherently ill-posed; however, with enough structural assumption on $\mathcal{X}$ and $\mathcal{N}$, it has been established that separation is possible. Depending on the application one might be interested in estimating only $X$, which is referred to as *denoising*, or in recovering both $X$ and $N$ which is referred to as *demixing*. Both demixing and denoising arise in a variety of important practical applications in the areas of signal/image processing, computer vision, and statistics Chen et al. (2001); Elad et al. (2005); Bobin et al. (2007); Candès et al. (2011).

Most of the existing techniques assume some prior knowledge on the structures of $\mathcal{X}$ and $\mathcal{N}$ in order to recover the desired component signal(s). Prior knowledge about the structure of $\mathcal{X}$ and $\mathcal{N}$ can only be obtained if one has access to the generative mechanism of the signals or has access to clean samples from the probability distribution defined over sets $\mathcal{X}$ and $\mathcal{N}$. In many practical settings, neither of these may be feasible. In this paper, we consider the problem of separating constituent signals from superposed observations when clean access to samples from the distribution is not available. In particular, we are given superposed observations $\{Y_i = X_i + N_i\}_{i=1}^{K}$ where $X_i \in \mathcal{X}$ and $Y_i \in \mathcal{N}$ are i.i.d samples from their respective (unknowns) distributions. In this setup, we explore two questions: First, *How can one learn prior knowledge about the individual components from superposition samples?* Second, *Can we leverage the implicitly learned constituent distributions for tasks such as denoising and demixing?*

### 1.1 SETUP AND OUR TECHNIQUE

Motivated by the recent success of generative models in high dimensional statistical inference tasks such as compressed sensing in Bora et al. (2017; 2018), in this paper, we focus on Generative Adversarial Network (GAN) based generative models to implicitly learn the distributions, i.e., generate

samples from their distributions. Most of the existing works on GANs typically assume access to clean samples from the underlying signal distribution. However, this assumption clearly breaks down in the superposition model considered in our setup, where the structured superposition makes training generative models very challenging.

In this context, we investigate the first question with varying degrees of assumption about the access to clean samples from the two signal sources. We first focus on the setting when we have access to samples only from the constituent signal class $\mathcal{N}$ and observations, $Y_i$'s. In this regard, we propose the *denoising*-GAN framework. However, assuming access to samples from one of the constituent signal class can be restrictive and is often not feasible in real-world applications. Hence, we further relax this assumption and consider the more challenging demixing problem, where samples from the second constituent component are not available and solve it using what we call the *demixing*-GAN framework.

Finally, to answer the second question, we use our trained generator(s) from the proposed GAN frameworks for denoising and demixing tasks on unseen test samples (i.e., samples not used in the training process) by discovering the best hidden representation of the constituent components from the generative models. In addition to the denoising and demixing problems, we also consider a compressive sensing setting to test the trained generator(s). **Below we explicitly list the contribution made in this paper:**

1. Under the assumption that one has access to the samples from one of the constituent component, we extend the canonical GAN framework and propose *denoising*-GAN framework. This learns the prior from the training data that is heavily corrupted by additive structured component. We demonstrate its utility in denoising task via numerical experiments.

2. We extend the above denoising-GAN and propose *demixing*-GAN framework. This learns the prior for both the constituent components from their superposed observations, without access to separate samples from any of the individual components. We demonstrate its utility in demixing task via numerical experiments.

The rest of this paper is organized as follows: In section 2, we discuss relevant previous works and compare our novelty over these existing methods. In section 3, we formally introduce our proposed approach, and in section 4, we provide some experimental results to validate our framework.

## 2 Application and Prior Art

To overcome the inherent ambiguity issue in problem 1, many existing methods have assumed that the structures of sets (i.e., the structures can be low-rank matrices, or have sparse representation in some domain (McCoy & Tropp, 2014)) $\mathcal{X}$ and $\mathcal{N}$ are a prior known and also that the signals from $\mathcal{X}$ and $\mathcal{N}$ are "distinguishable" (Elad & Aharon, 2006; Soltani & Hegde, 2016; 2017; Druce et al., 2016; Elyaderani et al., 2017; Jain et al., 2017). This assumption is a big restriction in many real-world applications. Recently, there have been some attempts to automate this *hard-coding* approach. Among them, structured sparsity Hegde et al. (2015), dictionary learning Elad & Aharon (2006), and in general manifold learning are the prominent ones. While these approaches have been successful to some extent, they still cannot fully address the need for prior structure. Over the last decade, deep neural networks have been demonstrated to learn useful representations of real-world signals such as natural images, and thus have helped us understand the structure of these high dimensional signals, for e.g. using deep generative models (Ulyanov et al., 2017).

In this paper, we focus on Generative adversarial networks GANs) Goodfellow et al. (2014) as the generative models for implicitly learning the distribution of constituent components. GANs have been established as a very successful tool for generating structured high-dimensional signals (Berthelot et al., 2017; Vondrick et al., 2016) as they do not directly learn a probability distribution; instead, they generate samples from the target distribution(s) (Goodfellow, 2016). Hence, they do not need to parametrize the distributions. If we assume that the structured signals are drawn from a distribution lying on a low-dimensional manifold, GANs generate points in the high-dimensional space that resemble those coming from the true underlying distribution.

Since their inception by Goodfellow et al. (2014), there has been a flurry of works on GANs (Zhu et al., 2017; Yeh et al., 2016; Subakan & Smaragdis, 2018). In most of the existing works on

GANs with few notable exceptions Wu et al. (2016); Bora et al. (2018); Kabkab et al. (2018); Hand et al. (2018); Zhu et al. (2016), it is implicitly assumed that one has access to clean samples of the desired signal. However, in many practical scenarios, the desired signal is often accompanied by unnecessary components. Recently, GANs have also been used for capturing of the structure of high-dimensional signals specifically for solving inverse problems such as sparse recovery, compressive sensing, and phase retrieval (Bora et al., 2017; Kabkab et al., 2018; Hand et al., 2018). Specifically, Bora et al. (2017) have shown that generative models provide a good prior to structured signals, for e.g., natural images, under compressive sensing settings over sparsity-based recovery methods. They rigorously analyze the statistical properties of a generative model based on compressed sensing and provide theoretical guarantees and experimental evidence to support their claims. However, they don't explicitly propose an optimization procedure to solve the recovery problem. They simply suggest using stochastic gradient-based methods in the low-dimensional latent space to recover the signal of interest. This has been addressed in Shah & Hegde (2018), where the authors propose using a projected gradient descent algorithm for solving the recovery problem directly in the ambient space (space of the desired signal). They provide theoretical guarantees for the convergence of their algorithm and also demonstrate improved empirical results over Bora et al. (2017).

While GANs have found many applications, most of them need direct access to the clean samples from the unknown distribution, which is not the case in many real applications such as medical imaging. AmbientGAN framework Bora et al. (2018) partially addresses this problem. In particular, they studied various measurement models and showed that their GAN can find samples of clean signals from corrupted observations. Although similar to our effort, there are several key differences between ours and AmbientGAN. Firstly, AmbientGAN assumes that the measurement model and parameters are known, which is a very strong and limiting assumption in real applications. One of our main contributions is addressing this limitation by studying the demixing problem. Second, for the noisy measurement settings, AmbientGAN assumes an arbitrary measurement noise and no corruption in the underlying component. We consider the corruption models in the signal domain rather than as a measurement one. This allows us to study the denoising problem from a highly structured corruption.

## 3 BACKGROUND AND THE PROPOSED IDEA

### 3.1 BACKGROUND

Generative Adversarial Networks (GANs) are one of the successful generative models in practice was first introduced by Goodfellow et al. (2014) for generating samples from an unknown target distribution. As opposed to the other approaches for density estimation such as *Variational Auto-Encoders (VAEs)* Kingma & Welling (2013), which try to learn the distribution itself, GANs are designed to generate samples from the target probability density function. This is done through a zero-sum game between two players, *generator*, $G$ and *discriminator*, $D$ in which the generator $G$ plays the role of producing the fake samples and discriminator $D$ plays the role of a cop to find the fake and genuine samples. Mathematically, this is accomplished through the following *min-max* optimization problem:

$$\min_{\theta_g} \max_{\theta_d} \quad \mathbb{E}_{x \sim \mathcal{D}_x}[log(D_{\theta_d}(x))] + \mathbb{E}_{z \sim \mathcal{D}_z}[log(1 - D_{\theta_d}(G_{\theta_g}(z)))], \tag{2}$$

where $\theta_g$ and $\theta_d$ are the parameters of generator networks and discriminator network respectively, and $\mathcal{D}_x$ denotes the target probability distribution , and $\mathcal{D}_z$ represents the probability distribution of the hidden variables $z \in \mathbb{R}^h$, which is assumed either a uniform distribution in $[-1, 1]^h$, or standard normal. it has been shown that if $G$ and $D$ have enough capacity, then solving optimization problem 2 by alternative stochastic gradient descent algorithm guarantees the distribution $\mathcal{D}_g$ at the output of the generator converges to $\mathcal{D}_x$. Having discussed the basic setup of GANs next we discuss the proposed modifications this to basic GAN setup that allow for usage of GANs as a generative model for denoising and demixing structured signals.

### 3.2 DENOISING-GAN

Our idea is inspired by AmbientGAN due to Bora et al. (2018) in which they used a regular GAN architecture to solve some inverse problems such as inpainting, denoising from unstructured noise,

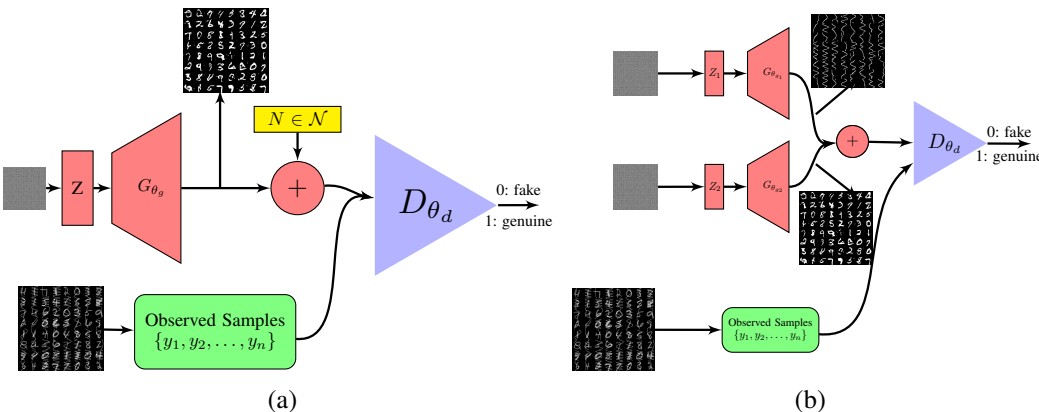

Figure 1: The architecture of proposed GANs. (a) denoising-GAN. (b) demixing-GAN.

etc. In particular, assume that instead of clean samples, one has access to a corrupted version of the samples where the corruption model is captured by a known random function $f_N(n)$ where $N$ is a random variable. For instance, the corrupted samples can be generated via just adding noise, i.e., $Y = X + N$. The idea is to feed Discriminator with observed samples, $y_i$'s (distributed as $Y$) together with the output of generator $G$ which is corrupted by model $f_N(n)$. Our denoising-GAN framework is illustrated in Figure 1(a). This framework is similar to one proposed in Ambient GAN paper however, the authors did not consider denoising task from structured superposition based corruption. In the experiment section, we show that denoising-GAN framework can generate clean samples even from structured corruption.

Now we use the trained denoising-GAN framework for denoising of a new test corrupted image which has not been used in the training process. To do that we use our assumption that our components have some structure and the representation of this structure is represented by the last layer of the trained generator, i.e., $X \in G_{\widehat{\theta}_g}$[1]. This observation together with this fact that in GANs, the low-dimension random vector $z$ is representing the hidden variables, leads us to this point: in order to denoise a new test image we have to find a hidden representation, giving the smallest distance to the corrupted image in the space of $G_{\widehat{\theta}_g}$ (Shah & Hegde, 2018; Bora et al., 2017). In other words, we have to solve the following optimization problem:

$$\widehat{z} = \arg\min_z \|u - G_{\widehat{\theta}_g}(z)\|_2^2 + \lambda \|z\|_2^2, \tag{3}$$

where $u$ denotes the corrupted test image. The solution of this optimization problem provides the (best) hidden representation for an unseen image. Thus, the clean image can be reconstructed by evaluating $G_{\widehat{\theta}_g}(\widehat{z})$. While optimization problem 3 is non-convex, we can still solve it by running gradient descent algorithm in order to get a stationary point[2].

### 3.3 DEMIXING-GAN

Now, we go through our main contribution, demixing. Figure 1(b) shows the GAN architecture, we are using for the purpose of separating or demixing of two structured signals form their super-position. As illustrated, we have used two generators and have fed each of them with a random noise vector $z \in \mathbb{R}^{100}$ uniformly distributed in $[-1,1]^{100}$. We also assume that they are independent of each other. We have used the same architecture for both of $G_{\theta_{g_1}}$ and $G_{\theta_{g_2}}$ where they are implemented as the generator of DCGAN (Radford et al., 2015). Next, the output of generators are summed up and the result is fed to the discriminator along with the superposition samples, $y_i's$. The architecture of the discriminator is also chosen according to the discriminator of DCGAN. In Figure 1(b), we just show the output of each generator after training for an experiment case in which the mixed image consists of 64 MNIST binary image LeCun & Cortes (2010) (for $X$ part) and a

---

[1] $G_{\widehat{\theta}_g}(.)$ denotes the trained generator network with parameter $\widehat{\theta}_g$.

[2] While we cannot guarantee the stationary point is a local minimum, but the empirical experiments show that gradient descent (implemented by backpropagation) can provide a good quality result.

second component constructed by random sinusoidal (for $N$ part) (please see the experiment section for the details of all experiments). Somewhat surprisingly, this architecture based on two generators can produce samples from the distribution of each component after enough number of training iterations. We note that this approach is fully unsupervised as we only have access to the mixed samples and nothing from the samples of constituent components is known. As mentioned above, this is in sharp contrast with AmbientGAN and our previous structured denoising approach. As a result, the demixing-GAN framework can generate samples from the second components (for example sinusoidal, which further can be used in the task of denoising where the corruption components are generated from highly structured sinusoidal waves). Now similar to the denoising-GAN framework, we can use the trained generators in Figure 1(b), for demixing of the constituent components for a given test mixed image which has not been used in training. Similarly, we have to solve the following optimization problem:

$$\widehat{z_1}, \widehat{z_2} = \arg\min_{z_1, z_2} \|y - G_{\widehat{\theta}_{g_1}}(z_1) - G_{\widehat{\theta}_{g_2}}(z_2)\|_2^2 + \lambda_1 \|z_1\|_2^2 + +\lambda_2 \|z_2\|_2^2, \tag{4}$$

where $u$ denotes the test mixed image and each component can be estimated by evaluating $G_{\widehat{\theta}_{g_1}}(\widehat{z_1})$ and $G_{\widehat{\theta}_{g_2}}(\widehat{z_2})$[3]. Similar to the previous case, while the optimization problem in 4 is non-convex, we can still solve it through block coordinate gradient descent algorithm, or in a alternative minimization fashion. We note that in both optimization problems 3 and 4, we did not project on the box set $[-1, 1]^{100}$. Instead we have used regularizer terms in the objective functions (which are not meant as projection step). We empirically have observed that imposing these regularizers can help to obtain good quality images in our experiment; plus, they may help that the gradient flow to be close in the region of interest by generators. This is also used in Bora et al. (2017).

## 4 NUMERICAL EXPERIMENTS

In this section, we present various experiments showing the efficacy of the proposed frameworks (depicted in Figure 1(a) and Figure 1(b)) in three different setups. First, we will focus on the denoising from structured corruption both in training and testing scenarios. Next, we focus on demixing signals from structured distributions. Finally, we explore the use of generative models from the proposed GAN frameworks compressive sensing setup. In all the following experiments, we did our best for choosing all the hyperparameters. The network architectures for discriminator and generator in these experiments are similar to ones proposed in DCGAN Radford et al. (2015). DCGAN is a CNN based GAN, in which there are some convolution layers used in both generator and discriminator followed by batch normalization (except the last layer of the generator and first layer of discriminator). All the nonlinear activation layers are Relu except the last layer of the discriminator. Due to the lack of space, we defer the details of experiments setup, the complementary experiments on the compressive sensing scenario, and experiments on the other dataset to the appendix.

### 4.1 STRUCTURED CORRUPTION MODELS

For all the experiments, we have used binary MNIST dataset. For the corruption part, we have used two structured noise models similar to Chen & Srihari (2014). In the first one, we generate random vertical and horizontal lines and add them to the dataset. The second structured noise is constructed based on random sinusoidal waves in which the amplitude, frequency, and phase are random numbers. We define the level of corruption as the number of sinusoidal or lines added to the original image. We note that both of these corruption models are highly structured. These two corruption model along with the clean binary MNIST image have been shown in the left panel of figure 2.

### 4.2 DENOISING FROM STRUCTURED CORRUPTION – TRAINING

We use GAN architecture illustrated in Figure 1(a) for removing of the structured noise. The setup of the experiment is as follows: we use $55000$ images with size $28 \times 28$ corrupted by either of the above corruption models[4]. The resulting images, $y_i$'s are fed to the discriminator. We also use

---

[3] $G_{\widehat{\theta}_{g_1}}(.)$ and $G_{\widehat{\theta}_{g_2}}(.)$ denote the first and second trained generator with parameter $\widehat{\theta}_{g_1}$ and $\widehat{\theta}_{g_2}$, respectively.

[4] We set level of corruption 1 for sinusoidal and 2 for the vertical and horizontal lines.

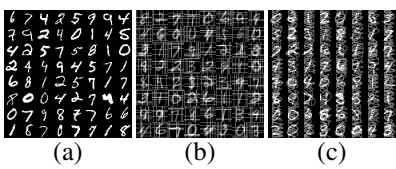 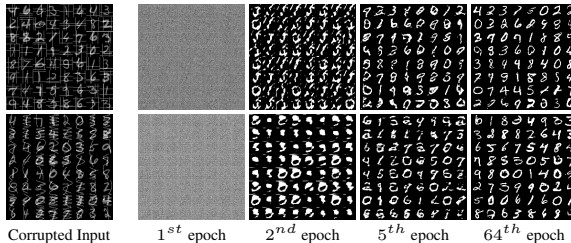

   (a)      (b)      (c)        Corrupted Input  $1^{st}$ epoch  $2^{nd}$ epoch  $5^{th}$ epoch  $64^{th}$ epoch

Figure 2: Left panel: (a). Clean binary MNIST image. (b). Corrupted image with random horizontal and vertical lines. (C). Corrupted image with random sinusoidal waves. Right panel: Evolution of outputted samples by generator for fixed $z$. Top row is for random horizontal and vertical corruption. Bottom row is for random sinusoidal corruption.

hidden random vector $z \in \mathbb{R}^{100}$ drawn from a uniform distribution in $[-1,1]^{100}$ for the input of the generator. During the training, we use 64 mini-batches along with the regular loss function in the GAN literature, stated in problem 2. The optimization algorithm is set to Adam optimizer, and we train discriminator and generator one time in each iteration. We set the number of epochs to 64. To show the evolution of the quality of output samples by the generator, we fix an input vector $z$ and save the output of the generator at different times during the training process. Figure 2 shows the denoising process for both corruption models. As we can see, the GAN used in estimating the clean images from structured noises is able to generate clean images after training of generator and discriminator. In the next section, we use our trained generator for denoising of new images which have not been used during the training.

### 4.3 Denoising from structured corruption – Testing

Now we test our framework with test corrupted images for both models of corruption introduced above. For reconstructing, we solve the optimization problem in 3 to obtain solution $\widehat{z}$. Then we find the reconstructed clean images by evaluating $G_{\widehat{\theta}_g}(\widehat{z})$. In the figures 3, we have used the different level of corruptions for both of the corruption models. In the top right, we vary the level of corruption from 1 to 5 with random sines. The result of $G_{\widehat{\theta}_g}(\widehat{z})$ has been shown below of each level of corruption. In the top left, we have a similar experiment with various level of vertical and horizontal corruptions. Also, we show the denoised images in the below of the corrupted ones. As we can see, even with heavily corrupted images (level corruption equals to 5), GAN is able to remove the corruption from unseen images and reconstruct the clean images.

In the bottom row of Figure 3, we evaluate the quality of reconstructed images compared to the corrupted ones through a classification test. That is, we use a pre-trained model for MNIST classifier which has test accuracy %98[5]. We feed the MNIST classifier with both denoised (output of denoising-GAN) and corrupted images with a different level of corruptions. For the ground truth labels, we use the labels corresponding to the images before corruption. One interesting point is that when the level of corruption increases the denoised digits are sometimes tweaked compared to the ground truth. That is, in pixel-level, they might not close to the ground truth; however, semantically they are the same. We also plot the reconstruction error per pixel (normalized by 16 images).

### 4.4 Demixing of structured signals – Training

In this section, we present the results of our experiments for demixing of the structured components. To do this, we use the proposed architecture in Figure 1(b). For the first experiment, we consider four sets of constituent components. In the first two, similar to the denoising case, we use both random sinusoidal waves and random vertical and horizontal lines for the second structured constituent component. The difference here is that we are interested in generating of the samples from the second component as well. In figure 4, we show the training evolution of two fixed random vectors, $z_1$

---

[5]The architecture comprises of two initial convolutional layers along with max-pooling followed by a fully connected layer and a dropout on top of it. Relu is used for all the activation functions and a soft-max function is used in the last layer.

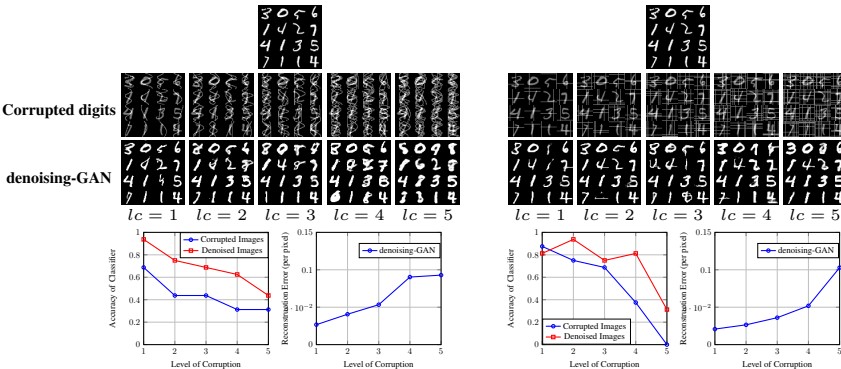

Figure 3: The performance of trained generator in different *level of corruption (lc)* for denoising of unseen corrupted digits. Top row: Ground truth digits together with corrupted digits with random sinusoids, and vertical and horizontal lines with a level of corruption from 1 to 5. Bottom row: Classification accuracy of pre-trained MNIST classifier for both corrupted and denoised digits along with the reconstruction error per pixel.

and $z_2$ in $\mathbb{R}^{100}$ in the output of two generators. In the left panel, we have used one random sinusoidal waves for the second component. As we can see, our proposed GAN architecture can learn two distributions and generate samples from each of them. In the right panel, we repeat the same experiment with random vertical and horizontal lines as the second component. While there is some notion of mode collapse, still two generators can produce the samples from the distribution of the constituent components.

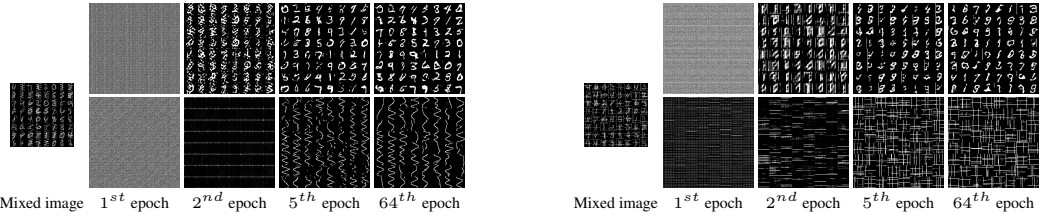

Figure 4: Evolution of output samples by two generators for fixed $z_1$ and $z_2$. The right panel shows the evolution of the two generators in different epochs where the mixed images comprise of digits and sinusoidal. The first generator is learning the distribution of MNIST digits, while the second one is learning the random sinusoidal waves. Left Panel shows the same experiment with random horizontal and vertical lines as the second components in the mixed images.

In the second scenario, our mixed images comprise of two MNIST digits from 0 to 9. In this case, we are interested in learning the distribution from which each of the digits is drawn. The left panel in Figure 5 shows the evolution of two fixed random vectors, $z_1$ and $z_2$. As we can see, after 32 epoch, the output of the generators would be the samples of MNIST digits. Finally, in the last scenario, we generate the mixed images as the superposition of digits 1 and 2. In the training set of MNIST dataset, there are around 6000 digits 1 and 2. We have used these digits to form the set of superposition images. The left panel of Figure 5 shows the output of two generators, which can learn the distribution of the two digits. The interesting point is that these experiments show that each GAN can learn the existing digit variety in MNIST training data set, and we typically do not see mode collapse, which is a major problem in the training of GANs (Goodfellow, 2016).

## 4.5 DEMIXING OF STRUCTURED SIGNALS – TESTING

Similar to the test part of denoising, in this section, we test the performance of two trained generators in a demixing scenario for the mixed images, which have not been seen in the training time. In Figure 6, we have illustrated demixing on three different input mixed images. In the left and middle panel, we consider the mixed images generated by adding a digit (drawn from MNIST test dataset) and a random sinusoidal. Then the goal is to separate (demix) these two from the given superimposed

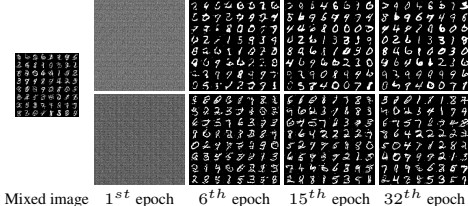 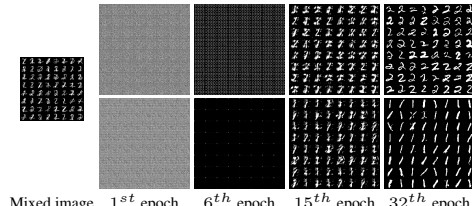

Mixed image  $1^{st}$ epoch  $6^{th}$ epoch  $15^{th}$ epoch  $32^{th}$ epoch          Mixed image  $1^{st}$ epoch  $6^{th}$ epoch  $15^{th}$ epoch  $32^{th}$ epoch

Figure 5: Evolution of output samples by two generators for fixed $z_1$ and $z_2$. The right panel shows that each generator is learning the distribution of one digit out of all 10 possible digits. The mixed images comprise two arbitrary digits between 0 to 9. Left Panel shows a similar experiment where the mixed images comprise only digits 1 and 2.

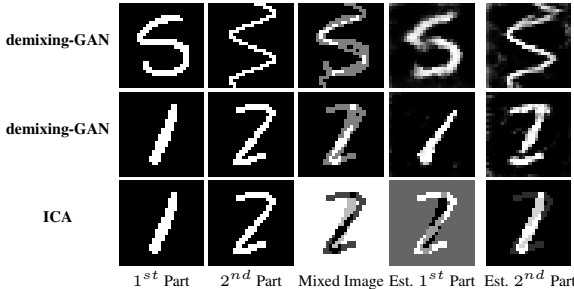

$1^{st}$ Part  $2^{nd}$ Part  Mixed Image  Est. $1^{st}$ Part  Est. $2^{nd}$ Part

Figure 6: The performance of trained generators for demixing of two constituent components. The first two columns are the ground-truth components. The third column is the ground-truth mixed image and the last two columns denote the recovered components. The first row uses the same generator trained for only one digit (drawn from MNIST test dataset) and a random sinusoidal. The third row uses the generator trained only for digits 1 and 2. The last row shows the result of demixing with ICA method.

image. To do this, we use GAN trained for learning the distribution of digits and sinusoidal waves (the left panel of Figure 4) and solve the optimization problem in 4 through an alternative minimization fashion. As a result, we obtain $\widehat{z_1}$ and $\widehat{z_2}$. The corresponding constituent components in each panel is then obtained by evaluating $G_{\widehat{\theta}_{g_1}}(\widehat{z_1})$ and $G_{\widehat{\theta}_{g_2}}(\widehat{z_2})$. Figure 6 shows our third experiment in which the input is the superposition of two different components. Here, we have compared the performance of demixing-GAN with *Independent component analysis (ICA)* method (Hoyer et al., 1999). The first two columns denote the ground-truth of the constituent components. The middle one is the mixed ground-truth, and the last two shows the recovered components using demixing-GAN and ICA. In the first two rows, the mixed image comprises of one digit, drawn from MNIST test dataset and a random sinusoidal. For this setup, we have used the trained GAN, described in the left panel of figure 4. In the last two rows, digits 1 and 2 drawn from the MNIST test dataset have been used for the constituent components where we have used the GAN trained for learning the distribution of digits 1 and 2 (right panel in figure 5). As we can see, our proposed GAN can separate two digits; however, ICA method completely fails in demixing of two components. In addition, Table 1 has compared numerically the quality of recovered components with the corresponding ground-truth ones through *mean square error (mse)* and *Peak Signal-to-Noise ratio (PSNR)* criteria.

**Remark:** Finally, as an attempt to understand the capability of demixing-GAN, we empirically observed that the hidden representation ($z$ space) of the generators for characterizing the distribution of the constituent components play an essential role to the success/failure of the demixing-GAN. We investigate this observation through some numerical experiment in section 5.5 in the appendix.

|  | MSE ($1^{st}$ Part) | MSE ($2^{nd}$ Part) | PSNR ($1^{st}$ Part) | PSNR ($2^{nd}$ Part) |
|---|---|---|---|---|
| First Row | 0.04715 | 0.03444 | 13.26476 | 14.62877 |
| Second Row | 0.04430 | 0.03967 | 13.53605 | 14.01344 |
| Third Row | 0.05658 | 0.05120 | 12.47313 | 12.90715 |
| Forth Row | 0.08948 | 0.10203 | 10.48249 | 9.91264 |

Table 1: Numerical Evaluation of the results in Figure 6 according to the *Mean Square Error (MSE)* and *Peak Signal-to-Noise ratio (PSNR)* criteria between the corresponding components in the superposition model.

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

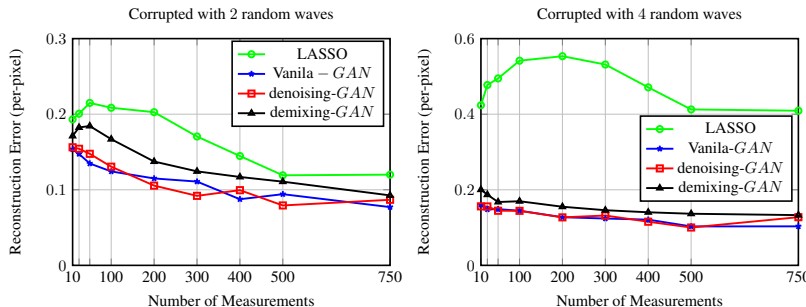

Figure 7: Performance of Different GANs in Compressive Sensing Experiments.

# 5    APPENDIX

## 5.1    SOME DETAIL INFORMATION FOR EXPERIMENTS

Here, we give some more details about the initialization of the generators we have used for the MNIST experiment in section 4.4. As we mentioned earlier, we have used the same architecture based on the DCGAN for both of the generators used in the demixing-GAN for MNIST dataset. In particular, the generators, comprise three layers: the first two layers are fully connected layers with Relu activation function. We have not used any batch-normalization (as opposed to the original DCGAN). Also, the weights in these fully connected layers are initialized according to the random normal distribution with standard deviation equals to 0.02. We have also used zero initialization for the biases. The third layer includes a transposed convolution layer with filters size of which are set to 5 and stride to 2. We have initialized these filters according to the random normal distribution with standard deviation equals to 0.02. Also, Relu nonlinearity has also been used after the transposed convolution operation. We have not used any max-pooling in this architecture. The discriminator also comprises two convolution layer with leaky-rely activation function followed by a fully connected layer without any max-pooling. The weights in these convolution layers have been initialized based on the truncated normal distribution with standard deviation equals to 0.02. The fully connected layer is initialized as before. Finally, we fed two generators with $i.i.d$ random noise vector with entries uniformly drawn from $[-1, 1]$.

## 5.2    COMPRESSIVE SENSING

In this section, we present results in the compressed sensing setting. In particular, we assume the following observation model by a random sensing matrix $A \in \mathbb{R}^{m \times p}$ with $m < p$ under structured corruption by $N$ as follows:

$$Y = A(X + N), \tag{5}$$

where the entries of $A$ are i.i.d Gaussian with zero mean and variance $\frac{1}{m}$. For a given generator model $G_{\widehat{\theta}}$, we solve the following inference problem:

$$\min_z \|Y - AG_{\widehat{\theta}}(z)\| + \lambda \|z\|_2^2, \tag{6}$$

where the generator $G_{\widehat{\theta}}$ is taken from various generator networks. For this experiment, we select the signal $X$ randomly from MNIST test dataset, and $N$ from random sinusoidal corruption with various number of waves. We compare generator models learned from clean MNIST images, i.e., generator model obtained from *denoising*-GAN and *demixing*-GAN framework. Both the generator models learned from denoising-GAN and demixing-GAN were trained under wave corruption model. This experiment tests the capability of the proposed GANs as the generative model for natural images. The experimental setup is similar to the reconstruction from Gaussian measurement experiment for MNIST data reported in Bora et al. (2017). For demixing-GAN, we select appropriate GAN by manually looking at the output of generators and choosing the one that gives MNIST like images as output. We also compare our approach to LASSO as MNIST images are naturally sparse. As reported in Bora et al. (2017). for this experiment the regularization parameter $\lambda$ was set as 0.1 as it gives the best performance on validation set. For solving the inference problem in 6, we have

used ADAM optimizer with step size set to $0.01$. Since the problem is non-convex, we have used $10$ random initialization with $10000$ iterations. We choose the one gives the best measurement error.

The results of this experiment are presented in Figure 7 where we report per-pixel reconstruction error on 25 images chosen randomly from the test dataset for two different corruption levels. The plot on the left panel is obtained when the signal is corrupted with $2$ random waves, whereas the right plot corresponds to the corruption with $4$ random waves. In the both figures, we observe that with corruption, the performance of LASSO significantly degrades. The generator from demixing-GAN improves the performance over LASSO. Generator obtained from denoising-GAN performs comparably to the vanilla GAN. This experiment establishes the quality of the generative models learned from GANs as the prior. The comparable performance of denoising GAN with vanilla GAN shows that meaningful prior can be learned even from heavily corrupted samples.

### 5.3 EXPERIMENTS ON F-MNIST DATASET

In this section, we illustrate the performance of the proposed demixing-GAN for another dataset, Fashion-MNIST (F-MNIST) (Han et al., 2017). This dataset includes 60000 training $28 \times 28$ grayscale images with 10 labels. The different labels denote objects, including T-shirt/top, Trouser, Pullover, Dress, Coat, Sandal, Shirt, Sneaker, Bag, and Ankle boot. Similar to the experiment with MNIST dataset being illustrated in Figure 5, we train the demixing-GAN where we have used InfoGAN Chen et al. (2016) architecture for the generators The architecture of the generators in InfoGAN is very similar to the DCGAN discussed above with the same initialization procedure. The dimension of input noise to the generators is set to $62$. We have also used the same discriminator in the MNIST experiment. Figure 8 shows the evolution of two fixed random vectors, $z_1$ and $z_2$ drawn from $[-1, 1]^{62}$. As we can see, after 21 epoch, the output of the generators would be the samples of F-MNIST objects. We also generate mixed images as the superposition of two objects, dress and bag images. In the training set of the F-MNIST dataset, there are around 6000 dress and bag images. We have used these images to form the set of superposition images. Figure 9 shows the output of two generators, which can learn the distribution of dress and bag images during 21 epochs.

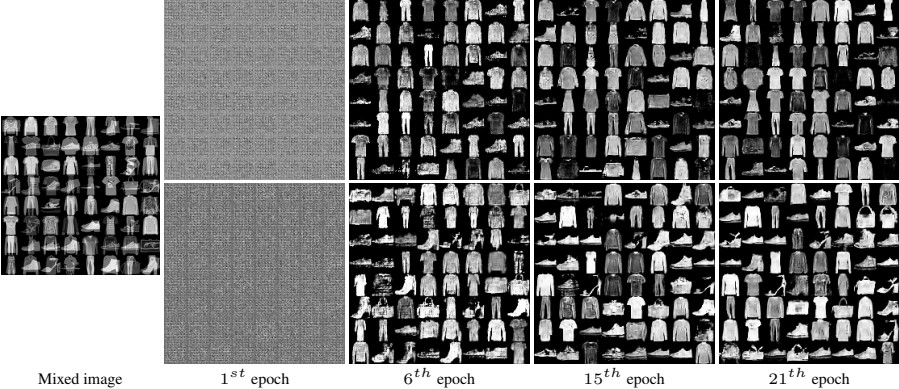

Figure 8: Evolution of output samples by two generators for fixed $z_1$ and $z_2$. The mixed images comprise two arbitrary objects drawn from 10 objects from training F-MNIST dataset. Each generator outputs the samples from the distribution of all 10 possible objects.

### 5.3.1 DEMIXING OF F-MNIST – TESTING

In this section, we evaluate the performance of trained demixing-GAN on the F-MNIST dataset. In Figure 10, we have illustrated an experiment similar to the setup in section 4.5. The first two columns in Figure 10 denote two objects from F-MNIST test dataset as the ground-truth components. The third column is the ground-truth mixed image, and the last two columns show the recovered constituent components. The first row uses the generator trained for only two objects for 20 epochs. The second row uses the generator trained for all 10 objects for 20 epochs. The third and fourth rows use the same generator trained for only two objects for 30 epochs. The last row shows the result of demixing with ICA method. We have implemented ICA using Scikit-learn module (Pedregosa et al., 2011). As we can see, ICA completely fails to separate the components (images of F-MNIST)

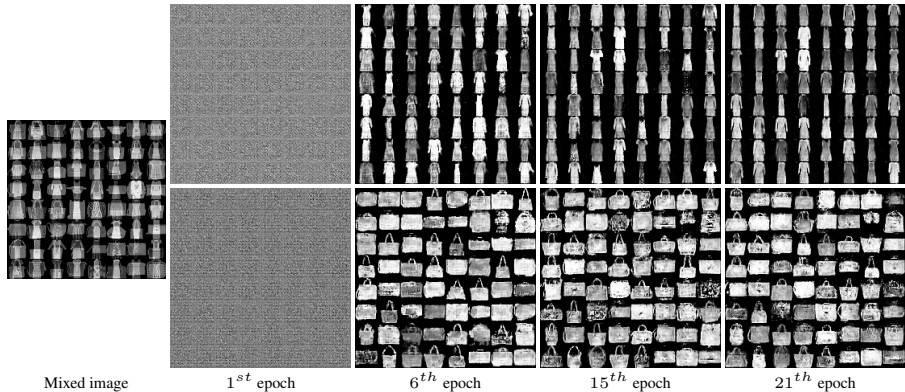

Figure 9: Evolution of output samples by two generators for fixed $z_1$ and $z_2$. The mixed images comprise only two objects, dress, and bag in training F-MNIST dataset. One generator produces the samples from dress distribution, while the other one outputs the samples from the bag distribution.

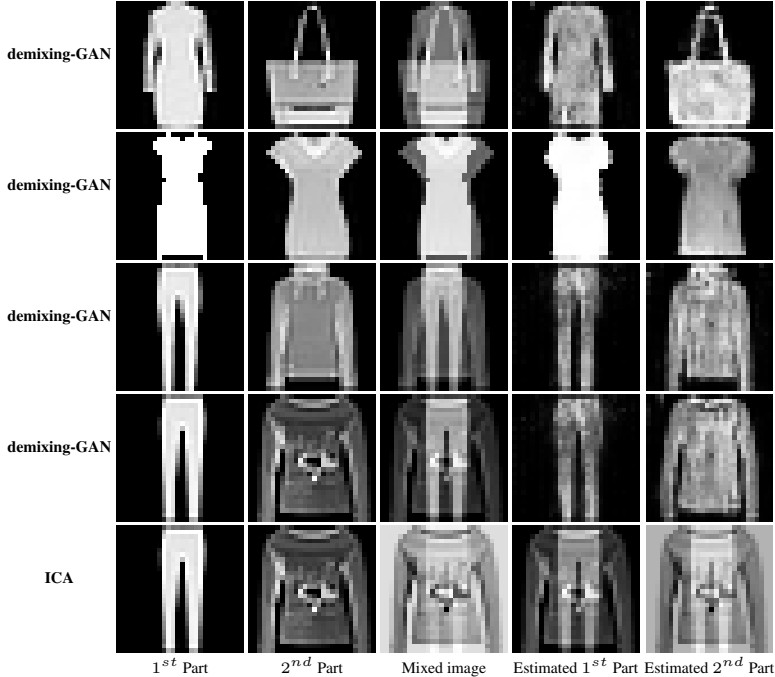

Figure 10: The performance of trained generators for demixing of two constituent components. The first two columns are the ground-truth components. The third column is the ground-truth mixed image and the last two columns denote the recovered components. The first row uses the generator trained for only two objects for 20 epochs. The second row uses the generator trained for all 10 objects for 20 epochs. The third and fourth rows use the same generator trained for only two objects for 30 epochs. The last row shows the result of demixing with ICA method.

from each other, while the proposed demixing-GAN can separate the mixed images from each other. However, the estimated image components are not exactly matched to the ground-truth ones (first two columns). This has been shown through numerical evaluation according to MSE and PSNR in Table 2.

## 5.4 EXPERIMENTS ON BOTH MNIST AND F-MNIST DATASET

In this section, we explore the performance of demixing-GAN when the superposed images comprise the sum of a digit $8$ from MNIST dataset and dress from the F-MNIST dataset. The experiment

| | MSE ($1^{st}$ Part) | MSE ($2^{nd}$ Part) | PSNR ($1^{st}$ Part) | PSNR ($2^{nd}$ Part) |
|---|---|---|---|---|
| First Row | 0.16859 | 0.12596 | 7.73173 | 8.99763 |
| Second Row | 0.05292 | 0.03304 | 12.76368 | 14.80992 |
| Third Row | 0.13498 | 0.11758 | 8.69732 | 9.29655 |
| Fourth Row | 0.12959 | 0.08727 | 8.87432 | 10.59132 |
| Fifth Row | 0.18250 | 0.12221 | 7.38733 | 9.12906 |

Table 2: Numerical Evaluation of the results in Figure 10 according to the *Mean Square Error (MSE)* and *Peak Signal-to-Noise ratio (PSNR)* criteria between the corresponding components in the superposition model.

for this setup has been illustrated in Figure 11. Since our goal is to separate dress from the digit $8$, for the first generator, we have used the InfoGAN architecture being used in the experiment in section 5.3 and similarly the DCGAN architecture for the second generator as section 4.4. As a result, the input noise to the first generator is drawn uniformly from $[-1, 1]^{62}$ and uniformly from $[-1, 1]^{100}$ for the second generator. Figure 11 shows the evolution of output samples by two generators for fixed $z_1$ and $z_2$. As we can see, after 21 epoch, the first generator is able to generate dress samples and the second one outputs samples of digit $8$.

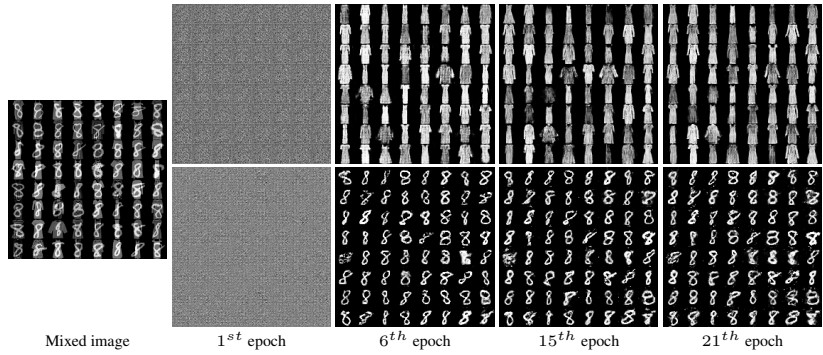

Mixed image   $1^{st}$ epoch   $6^{th}$ epoch   $15^{th}$ epoch   $21^{th}$ epoch

Figure 11: Evolution of output samples by two generators for fixed $z_1$ and $z_2$. The mixed images comprise only two objects, dress, and bag in training F-MNIST dataset. One generator produces the samples from digit $8$ distribution, while the other one outputs the samples from the dress distribution.

### 5.4.1 DEMIXING BOTH MNIST AND F-MNIST – TESTING

Similar to the previous Testing scenarios, in this section, we evaluate the performance of the demixing-GAN in comparison with ICA for separating a test image which is the superposition of a digit $8$ drawn randomly from MNIST test dataset and dress object drawn randomly from F-MNIST test dataset. Figure 12 shows the performance of demixng-GAN and ICA method. As we can see, ICA totally fails to demix the two images from each other, whereas the demixing-GAN is able to separate digit $8$ very well and to some extend the dress object from the input superposed image. MSE and PSNR values for the first component using ICA recovery method is given by $0.40364$ and $3.94005$, respectively. Also, MSE and PSNR for the first component using ICA recovery method is given by $0.15866$ and $7.99536$, respectively.

### 5.5 FAILURE OF THE DEMIXING-GAN

In this section, we empirically explore our empirical observation about the failure of the demixing-GAN in our setup. As we discussed briefly in section 4.5, here, we observe that if the hidden representation ($z$ space) of two generators are aligned to each other, then the two generators cannot output the samples in the signal space representing the distribution of the constituent components. To be more precise, in Figure 13, we consider separating digits $8$ and $2$ from their superpositions similar to the experiment in the right panel of Figure 5. However, here, we feed both generators with the same vector, i.e., $z_1 = z_2$ in each batch (this is considered as the extreme case where precisely the hidden variables equal to each other) and track the evolution of the output samples generated by both generators. As we can see even after 21 epochs, the generated samples by both generators are an unclear combination of both digits 2 and 8, and they are not separated clearly as

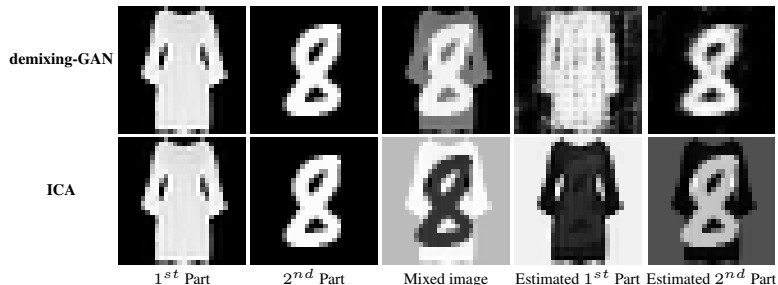

Figure 12: The performance of trained generators for demixing of two constituent components. The first two columns are the ground-truth components. The third column is the ground-truth mixed image and the last two columns denote the recovered components. The first row uses the generator trained through demixing-GAN. The second row shows the result of demixing with ICA method.

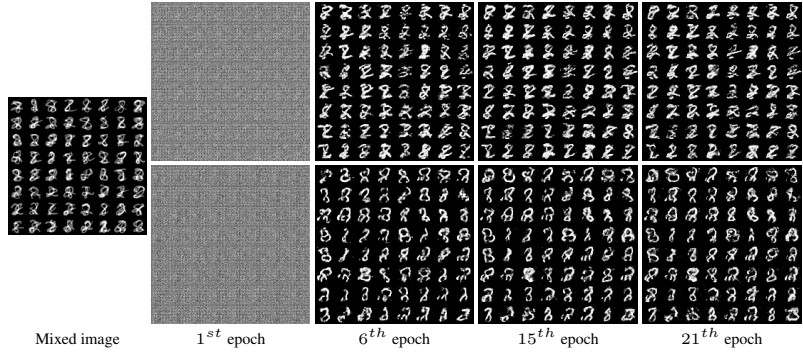

Figure 13: Failure of the demixing. Evolution of output samples by two generators for $z_1 = z_2$. The mixed images are the superposition of digits 2 and 8.

opposed to the case when we feed the generators with $i.i.d$ random vector $z$. We also repeat the same experiment with two aligned vectors $z_1$ and $z_2$, i.e., $z_2 = 0.1z_1$, Figure 14 shows the evolution of the output samples generated by both generators for this setup. As shown in this experiment, two generators cannot learn the distribution of digits 8 and 2. While we do not currently have a mathematical argument for this observation, we conjecture that the hidden representation ($z$ space) is one of the essential pieces in the demixing capability of the proposed demixing-GAN. We think that having (random) independent or close orthogonal vector $z$'s for the input of each generator is a necessary condition for the success of learning of the distribution of the constituent components and consequently demixing of them. Further investigation of this line of study is indeed an interesting research direction, and we defer it for the future research.

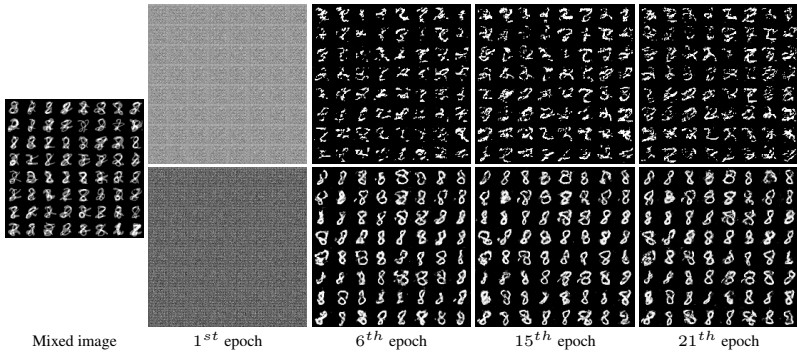

Figure 14: Failure of the demixing. Evolution of output samples by two generators for $z_1 = 0.1z_2$. The mixed images are the superposition of digits 2 and 8.

