# OpenReview forum: "LEARNING GENERATIVE MODELS FOR DEMIXING OF STRUCTURED SIGNALS FROM THEIR SUPERPOSITION USING GANS"
_ICLR.cc/2019/Conference_

### Official Review · AnonReviewer1 · 2018-11-01
**contribution, but experiments lacking**

**Rating:** 7
**Confidence:** 4

**Review:**

Quality is good, just a handful of typos.
Claritys above average in explaining the problem setting.
Originality: scan refs...
Significance: medium
Pros: the authors develop a novel GAN-based approach to denoising, demixing, and in the process train generators for the various components (not just inference). Further, for inference, the authors propose an explicit procedure. It seems like a noveel approach to demixing which is exciting.
Cons: The experiments do not push the limits of their method. It's difficult to judge the demixing 'power' of the method because it's difficult to tell how hard the problem is. Their method seems to easily solve it (super low MSE). The classification measure is clearly improved by denoising, which is totally unsurprising-- There should definitely be comparison with other denoising methods.

In general, they don't compare to any other methods. Actually in the appendix, comparisons are provided for a basic compressive sensing problem, but their only comparator is "LASSO" with a "fixed regularization parameter", and vanilla GAN. Since the authors "main contribution" (their words) is demixing, I'm surprised that they did not compare with other demixing approaches, or try on a harder problem. Could you  give some more details about the LASSO approach? How did you choose the L1 parameter?

I have another problem with the demixing experimental setting. On one hand, both the sinusoids and MNIST have "similar characteristics" in the sense that they are both pretty sparse, basically simple combinations of primary curves. This actually makes the problem harder for a dictionary learning approach like MCA (referenced in your paper). On the other hand, both signals are very simple to reconstruct. For example, what if you superimposed the grid of digits onto a natural image? Would you be able to train the higher resolution GAN to handle a more difficult setting? The other demixing setting of adding 1's and 2's has a similar problem.

The authors need to provide (R)MSE  results that show how well the method can reconstruct mixture components on average over the dataset. The only comparison is visual, and no comparators are provided.

Conclusions:
I'm actually torn on this paper. On one hand this paper seems novel and clearly contributes to the field. On the other hand, HOW MUCH contribution is not addressed experimentally, i.e. the method is not properly compared with other denoising or demixing methods, and definitely not pushed to its limits. It's hard to assess the difficulty of the denoising problem because their method does so well, and it's hard to assess the difficulty of demixing because of the lack of comparators.

Caveats:
I am knowledgeable about iterative optimization approaches to denoising and demixing, especially MCA (morphological component analysis), but *not knowledgeable about GAN-based approaches*, though I have familiarity with GANs.

*********************
Update after author response:
I think the Fashion-MNIST experiments and comparisons with ICA are many times more compelling than the original experiments. I think this is an exciting contribution to dually learning component manifolds for demixing.

---

> ### Author Response · Authors · 2018-11-26
> **We thank the reviewer 's encouraging feedback. We believe that the manuscript has been significantly strengthened by the reviewer's suggested changes.**
>
> We summarize the reviewer’s concerns below:
>
> The compressed sensing experiment was intended to test whether the proposed GAN based approaches can learn a generative model from heavily corrupted data. The experiment setup is similar to the one considered in the seminal paper Bora, et al (2016). "Compressed sensing using generative models." The parameters were chosen using the approach described in this paper. In the updated draft, we have provided more details about the LASSO approach about how the L1 parameter has been chosen and other details in section 5.2.
>
> -We tried to fix the typos and other grammar issues in the revised version.
>
> As the reviewer's' concern is similar to that of reviewer 2, we repeat our response here for the sake of completion:
>
> (1) We have also compared the quality of recovery of constituent components with the ground-truth through numerical criteria such as MSE and PSNR. Please see section 4.5 and 5.3.1 in the revised version of the paper. In the revised version of the paper, we have shown more experiments for another dataset, F-MNIST, and experiments based on both F-MNIST and MNIST datasets to support our proposed demixing-GAN. Please refer to appendix for the set of new experiments. In addition, we compare the performance of demixing-GAN with ICA method for both MNIST dataset, F-MNIST dataset, and a combination of them. Please see the appendix. As illustrated, ICA fails to demix superposed images from each other. We have to mention that while there are various methods for demixing of structured signals from their superpositions such as ones proposed by Hegde et al., 2012; Soltani and Hegde, 2017; McCoy and Tropp, 2014, these methods assume prior knowledge of the sets on which the constituent components lie. This is different from our setup in which there is prior knowledge is assumed and two generators in the demixing-GAN framework are responsible for providing this knowledge without hard-coding approach. As a result, we have selected only ICA as our benchmark to compare the performance of demixing-GAN.
>
> (2) Experiments with MNIST digits placed onto natural images is an interesting suggestion. We think these experiments are possible but given the time we are not able to do this experiment. However, we did an experiment in a similar spirit where we mixed images from F-MNIST and MNIST dataset. Our approaches were to successfully demix these two signals.
> Please see section 5.3 to 5.5 in the appendix of the revised version of the paper.

---

### Official Review · AnonReviewer3 · 2018-11-02
**Extension of AmbientGAN on denoising and demixing problems， but experiments are not sufficient**

**Rating:** 5
**Confidence:** 4

**Review:**

This paper proposed two new GAN structures for learning a generative modeling using the superposition of two structured components. These two structures can be viewed as an extension of AmbientGAN. Experiments results on MNIST dataset are presented. Overall, the demixing-GAN structure is relatively novel. However, the potential application seems limited and the experiment result is not sufficient enough to support the idea. Detail comments are as following,


1.	It seems there are no independent assumption imposed on the addition of two generators. It is possible that the possible model only will works on simple toy example, where the distributions of two structured components are drastic different. Or the performance will be affected by the initialization.  It would be nice if the author test this on more realistic examples, such as the source separation problem in acoustic or the unmixing problem in hyper-spectral images. More detail information about the experiments setting, such as the methods used to initialize the two generators are need.
2.	In the experiment part, it would be nice to have Quantitive results presented, for example PSNR for denoising. Simple comparison with several traditional methods could also help understanding the advantage of the model.

---

> ### Author Response · Authors · 2018-11-26
> **We thank the reviewer for his/her valuable feedback. We believe that the manuscript has been significantly strengthened by the reviewer's suggested changes.**
>
> We summarize the reviewer’s concerns below:
>
> Regarding (1):
> We agree that the demixing problem suffers from a fundamental separability issue which is sometimes referred to as incoherent of constituent components in the literature. This makes the demixing problem as an ill-posed problem. Please refer to the Answer (C) from the first reviewer for further discussion about this issue. In particular, please see section 5.5 in the appendix of the revised version of the paper where we have investigated an interesting empirical observation for the success/failure of the propose demixing-GAN approach based on the hidden variable space (z space) of the generators through some experiments.
>
> In the revised version, we have detailed more information about the experiments setting, such as the methods used to initialize the two generators for our experimental results in the appendix. We have added these details in section 5.1 of the revised version of the pape.
>
> Regarding (2):
> In the revised version of the paper, we have shown more experiments for another dataset, F-MNIST, and experiments based on both F-MNIST and MNIST datasets to support our proposed demixing-GAN. Please refer to appendix for the set of new experiments. In addition, we compare the performance of demixing-GAN with ICA method for both MNIST dataset, F-MNIST dataset, and a combination of them. Please see the appendix. As illustrated, ICA fails to demix superposed images from each other. We have to mention that while there are various methods for demixing of structured signals from their superpositions such as ones proposed by McCoy and Tropp (2014); Hegde et al. (2012); Soltani and Hegde, (2017), these methods assume prior knowledge of the sets on which the constituent components lie. This is different from our setup in which no prior knowledge is assumed and two generators in the demixing-GAN framework are responsible for providing the knowledge of the low-dimensional manifolds as opposed to the hard-coding approaches. As a result, we have selected only ICA as our benchmark to compare the performance of demixing-GAN.
>
> We have also compared the quality of recovery of constituent components with the ground-truth through numerical criteria such as MSE and PSNR. Please see section 4.5 and 5.3.1 in the revised version of the paper.

---

### Official Review · AnonReviewer2 · 2018-11-04

**Rating:** 4
**Confidence:** 5

**Review:**

In this, paper a GANs-based framework for additive (image) denoising and demixing is proposed. The proposed methodology for denoising largely relies on the Ambient GAN model and hence the technical contribution of the paper in this task appears to be limited. Regarding demixing, as explained in the comments below, the proposed model appears to be superficial in the sense that neither theoretical analysis nor thorough empirical evaluation is provided. The proposed method is evaluated on both tasks (i.e., denoising and demixing) by conducting toy experiments on handwritten digits (MNIST).

More specifically, the authors employ the Ambient GAN to train a generator that generates clean samples when the type of corruption is known (i.e., when corruption is modelled by a known function which interacts with the clean data in an additive way). For denoising, the authors propose to learn the latent variable that generates the clean test image by solving a ridge regularized non-convex inverse problem (Eq. 3). The problem is solved via gradient descent and theoretical analysis on the converge of the algorithm is not provided.  Clearly, this approach has limited practical applications since the corruption function needs to be known which rarely happens in practice.

Next, considering additive demixing, the authors assume that the corruption/structured signal is unknown but it can be modelled using a convolutional network (using the architecture of DCGAN). They employ the same network architecture for modelling the clean data generation process and learn the parameters of both generators using adversarial training. Demixing is performed by solving a similar by solving a similar ridge regularized non-convex inverse problem as in the case of denoising (i.e., Eq. 4). As authors mention in the paper, it is indeed surprising that the proposed GANs-based model with two generators is able to produce samples from the distribution of each signal component by observing only additive mixtures of these signals. Without any assumptions, the proposed model is not identifiable. This is my main concern regarding this paper and a theoretical investigation is definitely needed. My main questions revolve around under what conditions the column spaces of the two generators are mutually independent and what is the type of components structure that the proposed model can recover.

As mentioned above, the experimental evaluation is limited to the NMIST dataset while comparisons with existing related models such as RPCA and ICA that work efficiently and with guarantees in the additive setting studied in this paper are considered essential in order to prove empirically the merits of the proposed framework.

---

> ### Author Response · Authors · 2018-11-26
> **We thank the reviewer's valuable feedback. We believe that the manuscript has been significantly strengthened by the reviewer's suggested changes.**
>
> We summarize the reviewer’s concerns below:
> -The limited set of experiments
> -Lack of theoretical analysis for the proposed optimization problems
> -Identifiability issue in the demixing problem
> -Comparison with other methods
>
>
> Answer to (A):
> We acknowledge that the experiment only on MNIST dataset is limited and may not be very satisfactory. In the revised version, we have added the experiments for Fashion-MNIST (F-MNIST) dataset regarding our demixing setup to the appendix of the paper. Please see section 5.3 in the appendix for the details of this new set of experiments. The F-MNIST dataset includes 60000 training 28X28 gray-scale images with 10 labels for the objects. As discussed in the appendix, we have trained the demixing-GAN and used the trained generators for the demixing task for different experiment scenarios. In addition, We have conducted another set of experiments based on the mixing of MNIST digits with the F-MNIST objects. In this case, the proposed demixing-GAN is also able to learn the samples of the two components.
>
>
> Answer to (B):
> We acknowledge that the proposed demixing strategy (regularized optimization problem) has not been supported by theoretical guarantee. This is certainly an interesting future direction for us. In this current work, our goal is mostly to verify the ability of GANs for capturing the prior knowledge about the manifolds on which the constituent component lie through the numerical exploration. Since the optimization problem even for simpler denoising case is non-convex, its convergence analysis even for the stationary point is a very challenging problem. As our best knowledge, most of GANs paper suffer from this regard and most of them explore the properties of GAN framework through some numerical verification.
>
>
> Answer to (C):
> As the reviewer has mentioned the inherent identifiability of the constituent components makes the demixing problem an ill-posed problem. We are well-aware about this issue. Without some notion of so-called incoherent between the constituent components, separating of the components is not possible. This issue has been addressed in the Elad et al. (2005); Hegde et al. (2012). However, in these works, the structure of the components is assumed to be sparse in some known domain and through this assumption, the authors characterize the notion of incoherent. What GAN is generating can be considered as an appropriate prior knowledge about the manifolds on which the components lie.
>
> As an attempt to understand the capability of the demixing-GAN, we empirically observed that the hidden representation (z space) of the generators for characterizing the distribution of the components play an essential role to the success/failure of the demixing-GAN. We investigate this observation through some numerical experiment in section 5.5 in the appendix of the revised version. Through some empirical observation, we conjecture that having (random) independent or close orthogonal vector z's for the input of each generator is a necessary condition for the success of learning of the distribution of the constituent components and consequently demixing of them.
>
> We do not quite follow when the reviewer has mentioned that the “column spaces of two generators…”. GAN is a highly nonlinear and nonconvex map from low-dimensional (hidden variable space) to high-dimensional (signal space). We don’t think so that we can talk about the incoherent of the column space for the output of generators as their action on a vector z cannot be captured by a matrix-vector multiplication. But certainly, we agree with the reviewer about some notion of incoherent should be analyzed for fundamental identifiability issue between the components. This is the subject of our future study.
>
>
> Answer to (D):
> To address the lack of sufficient comparisons, we have provided some experiments through ICA for demixing of two digits from MNIST and F-MNIST datasets. As illustrated in the revised paper, ICA fails to separate the components from each other, while this is not the case for the proposed demixing-GAN. Regarding RPCA comment, if the reviewer means robust principal component analysis, we do not think that that RPCA is related to our setup. In RPCA, we assume that one part is sparse and another is low-rank. Posing the low-rank and sparse constraints in the output of generators is not clear. In this paper, we are mainly demonstrating that the generative models for two data sources can be learned from their additive superposition. Whereas, in RPCA, the structure of the two constituents signals is hard-coded and a fixed apriori. Therefore two approaches operate in fundamentally different settings. We agree that the comparison of the proposed model with RPCA seems reasonable and it is an interesting research question to pose the low-rank and sparse constraints in the output of generators. More investigation about this direction might be an interesting future research.

---

### Meta-Review · Area_Chair1 · 2018-12-19
**Interesting and promising novel approach for demixing, but with no theoretical grounding and limited experimental evaluation**

**Confidence:** 3
**Recommendation:** Reject

**Metareview:**

The paper proposes two simple generator architecture variants enabling the use of GAN training for the tasks of denoising (from known noise types) and demixing (of two added sources). While the denoising approach is very similar to AmbientGAN and could thus be considered somewhat incremental, all reviewers and the AC agree that the developed use of GANs for demixing is an interesting novel direction. The paper is well written, and the approach is supported by encouraging experimental results on MNIST and Fashion-MNIST.
Reviewers and AC noted the following weaknesses of the paper: a) no theoretical support or analysis is provided for the approach, this makes it primarily an empirical study of a nice idea.
b) For an empirical study, the experimental evaluation is very limited, both in terms of dataset/problems it is tested on; and in terms of algorithms for demixing/source-separation that it is compared against.
Following these reviews, the authors added the experiments on Fashion-MNIST and comparison with ICA which are steps in the right direction. This improvement moved one reviewer to positively update his score, but not the others.
Taking everything into account, the AC judges that it is a very promising direction, but that more extensive experiments on additional benchmark tasks for demixing and comparison with other demixing algorithms are needed to make this work a more complete contribution.